# How to Measure Foot Self-Care? A Methodological Review of Instruments

**DOI:** 10.3390/jpm13030434

**Published:** 2023-02-28

**Authors:** Jenni Sipilä, Anne-Marie Mäkelä, Sasu Hyytiä, Minna Stolt

**Affiliations:** 1Department of Nursing Science, University of Turku, 20520 Turku, Finland; 2Department of Nursing Science, Turku University Hospital, University of Turku, 20520 Turku, Finland; 3Pihlajalinna Pikku Huopalahti, 00300 Helsinki, Finland; 4Department of Nursing Science, University of Eastern, 70211 Kuopio, Finland

**Keywords:** foot, foot self-care, instrument, measurement, methodological review

## Abstract

Foot self-care is an important element of caring for and promoting foot health. However, little is known about the validity and reliability of existing foot self-care instruments. The purpose of this review is to describe and analyze the focus, content, and psychometric evidence of existing instruments for measuring foot self-care. A methodological review of three international scientific databases—Medline (PubMed), CINAHL (Ebsco), and Embase—was conducted in May 2022. The search produced 3520 hits, of which 53 studies were included in the final analysis based on a two-phase selection process. A total of 31 instruments were identified, of which six were observed to have been used more than once. Subsequently, the methodological quality of these six instruments was evaluated. It is noted that although a considerable variety of instruments are used in measuring foot self-care, only a small proportion are used consistently. In general, the psychometric testing instruments seem to primarily focus on analyzing content validity and homogeneity. In the future, comprehensive testing of instrument psychometrics could enhance the cumulative evidence of the methodological quality of these instruments. Furthermore, researchers and clinicians can use the information in this review to make informed choices when selecting an instrument for their purposes.

## 1. Introduction

Foot self-care is an important element of caring for and promoting foot health. Active and proper foot self-care help maintain foot health and prevent foot problems. Notably, foot problems are particularly prevalent in older people [1,2] and people with long-term health problems [3,4,5,6], such as diabetes mellitus and rheumatic conditions, thus emphasizing the urgent need for implementing preventive foot self-care. In this context, it is necessary to have valid instruments for evaluating patients’ foot self-care. However, a systematic summary of existing foot self-care instruments is lacking in the literature.

Self-care, in general, has become a central component of health care and patients’ own resources in terms of caring for, maintaining, and promoting their own health [7]. Foot self-care is a demanding daily task that entails maintaining fine motor skills, general mobility, and upper limb dexterity with sufficient muscle strength [8]. Although a universally agreed-upon definition of foot self-care is lacking, this review defines it as individuals’ own activities directed at caring for their own feet, including skin and nail care, foot pain, and use of proper footwear [9]. Foot self-care is important in caring for different foot disorders, such as dry skin, flatfoot, hallux valgus, and metatarsalgia. To conduct foot self-care, it requires competence [10], including personal knowledge, skills, motivation, and physical ability. In this context, proper knowledge relates to evidence and good practice guidelines about foot self-care. Furthermore, skills refer to individuals’ abilities to conduct activities, such as skin moisturisation or nail cutting, according to evidence-based guidelines and recommendations. In addition, motivation is also significant for regularly caring for one’s feet.

Adherence to foot self-care among people/patients is diverse. Although foot self-care is considered important by patients, it is generally conducted unsystematically [8]. In fact, the importance of preventive foot self-care is often recognized when foot problems have already occurred [11]. Poor or improper foot self-care can, in turn, negatively affect one’s foot health status. For example, the foot self-care activities of patients with diabetes who suffer from foot ulcers have been demonstrated to be improper [12,13]. Therefore, a systematic measurement of patients’ foot self-care is required to identify potential gaps in their competence.

The evaluation of the validity and reliability of foot self-care instruments is a constant process, as they are used in various contexts and populations. Therefore, cumulative evidence is needed to prove the validity and reliability of a particular instrument. A single study can only provide evidence of the psychometrics in a certain sample and, therefore, cannot be regarded as the only source of evidence. This indicates that robust reporting of the instruments’ development process and the results of their psychometric testing are needed to gather relevant evidence of their validity and reliability. Moreover, instruments used as self-reported outcome tools are useful only if there is evidence to support the interpretation of their obtained scores [14]. Therefore, accurate interpretations of reliability and validity considering different settings and samples can be made only when such kinds of necessary information are available. Furthermore, this information aids researchers in understanding the instrument development process, testing, and its results. From a clinical point of view, such information provides an opportunity to use instruments that have been consistently assessed as reliable.

The validity and reliability of foot- and ankle-related instruments have increasingly been under investigation. However, the target of most studies has been the assessment of foot and ankle symptoms and functions rather than foot self-care. For example, recent reviews have prominently focused on patient-reported outcome measures (PROMs) in foot and ankle orthopaedics to reveal variability and deficits in methodological quality and instrumentation [15,16] and highlighted many studies that have used non-validated instruments [17] with limited evidence of their psychometric properties [18]. Similarly, a substantial variability in the measurement properties of instruments assessing foot-related disabilities was identified in the studies focusing on patients with rheumatoid arthritis [19,20]. In contrast, foot- and ankle-related studies conducted on people with diabetes mellitus seemed to maintain a sufficient level of methodological rigour and used valid instruments to measure issues, such as diabetic neuropathy [21]. However, a review focusing on foot self-care instruments still seems to be lacking.

Previous reviews on the impact of foot care education on foot self-care have criticized the heterogeneity of the available assessment tools [22,23]. These instruments focused on measuring changes in knowledge levels, foot care behaviour, and foot health [23]. Moreover, they were developed for the purposes of a single study and measured the technical competence to carry out a certain skill or regularity of desired foot care behaviour rather narrowly [23]. Furthermore, the content areas that the instruments focus on and their validity remain unclear. Furthermore, the lack of validated outcome tools hampers the precision of the instruments’ measurements and their ability to measure changes in behaviour, thus revealing flaws in their evaluation of the impact of foot care education.

Poorly conducted foot self-care can increase the risk of foot complications [24] and seriously decrease the level of foot health [25]. Thus, the proper evaluation of foot self-care requires valid and reliable instruments. To address this challenge, a methodological psychometric review of existing foot self-care instruments could promote and facilitate their use in clinical practice and research. Therefore, the purpose of this review is to describe and analyse the focus, content, and psychometric evidence of existing instruments that measure foot self-care. The ultimate goal is to gather and provide information regarding the accurate assessment of foot self-care and the measurement properties of individual instruments.

Therefore, the primary questions that this review seeks to answer are as follows:(1)What is the focus and content of the instruments that measure foot self-care?(2)What is the psychometric evidence for these foot self-care instruments?

## 2. Materials and Methods

### 2.1. Design

This study was conducted by applying a methodological review design. The reporting process was carried out in accordance with the Preferred Reporting Items for Systematic Reviews and Meta-Analyses (PRISMA) statement [26]. Moreover, a review protocol was planned, but not published, prior to conducting the review.

### 2.2. Eligibility Criteria

The identified studies were included in the sample for this investigation if they: (1) Were an empirical primary study with a focus on foot self-care; (2) Included an instrument that measures foot self-care (subjective or objective); (3) Provided evidence of psychometric properties of the foot self-care instrument; and (4) Written in English. The exclusion criteria for the studies included (1) Theoretical discussion papers or (2) The use of general self-care instruments.

### 2.3. Information Sources and Search Strategy

A methodological review was conducted across three international scientific databases (Medline (PubMed), CINAHL (Ebsco), and Embase) in May 2022. The search sentence was ((foot[Title/Abstract]) AND (self[Title/Abstract])) AND (care[Title/Abstract] OR caring[Title/Abstract] OR manage[Title/Abstract] OR management[Title/Abstract] OR efficacy[Title/Abstract]). This search was limited to the title and abstract levels and to studies written in English. Furthermore, no time limit was applied. Moreover, although the review protocol was planned a priori, it was neither published nor registered.

### 2.4. Selection Process

The process of selecting the relevant studies included screening the records and evaluating their eligibility against the inclusion and exclusion criteria (Figure 1). All duplicate records were excluded in the screening phase. Following this, the titles and abstracts of the studies were inspected by two independent researchers (A-MM, MS). After reaching a consensus, the full texts of the included studies were read and evaluated. After each step, the researchers discussed their selections to ultimately reach a consensus. In cases of disagreement, a third researcher was consulted.

### 2.5. Data Collection Process and Data Items

A spreadsheet was developed particularly for the purposes of this review. It included the following information: author, year of publication, country of origin, aim of the study, name of the instrument, measurement focus, number of items, response options, and a list of studies using the particular instrument. While retrieving the data, the study authors’ original expressions for the instruments were used without inducing any additional interpretation.

### 2.6. Quality Appraisal

The Mixed Methods Appraisal Tool (MMAT, version 2018) [27] was used to evaluate the methodological quality of the selected studies. The MMAT consists of seven items: two general and five design-related items. The response to each item was registered in terms of a three-point scale (yes, no, and can’t tell).

### 2.7. Synthesis of Results

The instruments used by patients to measure their performed activities related to foot self-care were first identified from the original articles. After identification, they were listed and classified into groups according to their names. Notably, some of these instruments were used in multiple studies, while the revised versions of some were also reported.

Subsequently, the articles were categorized into studies that (a) Reported original instrument development research; (b) Reported further validation of a certain instrument; and (c) Used instruments without considering any information regarding their validation or psychometric testing. Based on this categorization, instruments that were used in multiple studies were identified, and further analysis was conducted on them.

Descriptive information of each instrument on the item level was gathered to a separate table. Items were grouped based on their characteristics to show which foot self-care content areas were present in these instruments. To have an understanding of item coverage, the number of items describing a certain aspect of foot self-care was summed up.

The psychometric properties of each instrument were analyzed using the framework proposed by Zwakhalen and colleagues [28], which includes 10 items that cover the most crucial aspects of validity and reliability: (1) Known origin of the items; (2) Sufficient sample for testing (number of participants); (3) Analysis of, and justification for, content validity; (4) Level of criterion validity achieved using correlation; (5) Construct validity in relation to other appropriate knowledge tests; (6) Construct validity of differentiation; (7) Homogeneity; (8) Inter-rater reliability (confirmed through observation or noted in activity); (9) Intra-rater or test–retest reliability; and (10) Feasibility. Each item was scored as either 0, 1, or 2 by the two researchers according to the relevant scoring criteria [28]. To gather an overall level of psychometric evidence, the scores were summed up. On summing up the scores, the maximum score stood at 20, with a higher score representing a higher level of the analyzed psychometric property. Originally, these criteria were created to evaluate a pain assessment tool for people with memory disorders [28]. Although the psychometric properties assessed in the original study are universal, the assessment criteria of this study were modified to correspond to foot self-care content. Moreover, the criteria suggested by Zwakhalen and her colleagues [28] have been previously used to assess the psychometric properties of instruments [29]. Notably, this framework was developed based on the methodological literature as a quality judgement criterion for instruments in nursing and health research.

## 3. Results

### 3.1. Study Selection

The search produced a total of 3520 hits (*n* = 1449 for Medline and PubMed, *n* = 466 for CINAHL and *n* = 1605 for Embase). After removing duplicates, 2417 hits were included in the study selection phase. Subsequently, 119 studies were selected after screening the titles and abstracts of the studies. Following this, based on the full texts of the selected studies, 53 that met the eligibility criteria were included in the final analysis.

### 3.2. Study Characteristics

A total of 31 instruments used in 53 studies were identified for further analysis (Appendix A). These instruments were observed to predominantly measure self-reported foot self-care behaviours or activities. The number of items in the instruments ranged from 4 to 29, while a five-point response scale was primarily used to indicate the frequency of activities related to foot self-care. Out of these 31 instruments, 25 were used only once or were developed for the purpose of a single study. In addition, six instruments were unnamed. Therefore, the analysis of psychometric properties targeted the instruments (*n* = 6) that were used more than once.

### 3.3. Description of the Analysed Instruments

This section provides a detailed description of the six instruments that were employed in multiple studies considered in the sample selected for this investigation.

The Diabetes Foot Self-care Behavior Scale (DFSBS) [30] measures the frequency of foot self-care behaviour. It entails seven items: checking the bottom of the feet and between toes, washing between toes, drying between toes after washing, applying lotion, inspecting the insides of shoes, and breaking in new shoes. Furthermore, this scale has two parts. Part 1 assesses the number of days that a respondent performs a certain behaviour during a 1-week period using the five-point scale (0 days, 1–2 days, 3–4 days, 5–6 days, 7 days). Meanwhile, Part 2 evaluates the frequency at which a respondent performs a certain foot self-care behaviour (5-point scale from 1 = never to 5 = always). These ratings are summed up to arrive at a score, with higher scores indicating a better performance of foot self-care behaviour [30]. This section may be divided by subheadings. It should provide a concise and precise description of the experimental results, their interpretation, as well as the experimental conclusions that can be drawn.

The Summary of Diabetes Self-Care Activities (SDSCA) [31] is a self-report instrument dealing with diabetes self-management. It covers all the self-care areas related to diabetes: general diet, specific diet, exercise, blood-glucose testing, foot care, and smoking. Since specific parts of this instrument can be used separately, many studies investigated in this review were observed to implement the foot care section of the SDSCA, which consists of five items that help to identify the number of days in a week that a person has performed diabetes foot self-care activities: feet washing, feet soaking, drying between the toes after washing, foot checks, and footwear inspection. The response to each item is registered on a scale from 0 to 7, based on the number of days that the person performed the activity (the higher the mean, the better the care) [31].

The Nottingham Assessment of Functional Footcare (NAFF) [32] is a self-report instrument for assessing the foot care behaviour of people with diabetes. This tool accounts for a total of 29 items to measure the extent to which people comply with recommended foot care behaviours. Response options for this instrument are provided on a four-point scale ranging from “rarely” to “most of the time.” The sum of the items’ scores is then calculated, with a higher score indicating better foot self-care [32].

The Diabetic Foot Self-Care Questionnaire of the University of Malaga, Spain (DFSQ-UMA) [33] was formulated to evaluate foot self-care among patients with diabetes. It consists of 16 questions that are divided into three domains: personal self-care, podiatric care, and shoes and socks. Each item is scored on a five-point response scale (1 = very inadequate; 5 = very adequate), while some items explore the frequency of a determined self-care activity (1 = never; 5 = always) [33].

The Foot Self-Care Behaviour Questionnaire [34] measures the frequency of performing foot care behaviour based on 17 items divided into two subscales: preventive foot self-care (nine items) and potentially foot-damaging behaviour (eight items). Similarly, two different response scales are used: a six-point scale (i.e., twice a day, daily, every other day, twice a week, once a week, or never) and a four-point scale (i.e., always, most of the time, occasionally, or never). The responses are summed up, wherein a higher score indicates more preventive and potentially damaging behaviours.

The Foot Self-Care Observation Guide (FSCOG) [35] is an objective observation measurement for detecting foot self-exam components. It consists of 16 items divided into three categories: foot care (five items), foot check (three items), and foot safety (eight items). The responses are provided on a five-point scale (1 = never, 2 = occasionally, 3 = sometimes, 4 = frequently, and 5 = always), and then the responses are summarized (range 15–75) with a higher score indicating better foot self-care behaviour.

### 3.4. Methodological Quality of the Included Studies

The methodological quality of the selected studies, as assessed by the MMAT [27], was found to be acceptable (Appendix A). However, the main deficit observed in the descriptive quantitative studies was related to non-response bias. Furthermore, the blinding of the assessors was seldom achieved in the randomized controlled trials. For the non-randomized studies, the main shortcoming was recognized as accounting for the confounders of the analysis.

### 3.5. Focus and Content of the Instruments

The selected instruments were implemented for primarily measuring the frequency of foot self-care (DFSBS [30]; SDSCA [31]; DFSQ-UMA [33], foot self-care activities (DFSQ-UMA [33]) and compliance with recommended foot care (NAFF [32]).

At the item level (Table 1), the focus was predominantly on the selection of the type of footwear (*n* = 15), followed by questions related to socks (*n* = 8), footwear assessment (*n* = 7), foot inspection (*n* = 7), drying the feet after washing (*n* = 6), and skin care (*n* = 6). A few other items that were targeted for evaluation include walking barefoot (*n* = 4), skin moisturization (*n* = 4), foot washing (*n* = 3), nail cutting (*n* = 3), attitude towards foot self-care (*n* = 2), and foot soaking (*n* = 1).

### 3.6. Psychometric Evidence of the Instruments

The evidence on the psychometric properties of the instruments varied (Table 2, Appendix A). First, it should be noted that all the assessed foot self-care instruments were developed based on comprehensive literature reviews. In addition, some authors (NAFF [32] and FSCOG [35]) incorporated foot care recommendations or guidelines to strengthen the theoretical basis of their instruments.

The DFSBS [30] was originally tested comprehensively on a sample of 295 patients with diabetes. Evidence of its content validity, construct validity, differentiation, and internal consistency was also provided. In addition, its feasibility was tested using a pilot test [30]. Over time, the DFSBS has been translated into Arabic [38], Iranian [36], Malay [39,41], and Turkish [40] languages. Moreover, several studies that provide evidence of the instrument’s construct validity and internal consistency [36,38,39,40,41] have been conducted. However, no evidence of its criterion or construct validity was provided, while its reliability testing focused only on homogeneity. Notably, its feasibility was evaluated by a few studies in pilot testing [30,38,39].

Limited reports could be acquired on the psychometric evidence for the SDSCA [31]. Only one study reported evaluating the feasibility of the instrument through a pilot study [45]. Meanwhile, it was noted that the NAFF [47] was tested for content validity and homogeneity [49].

The DFSQ–UMA [33] was found to be based on a comprehensive literature review followed by careful item operationalization. In addition, extensive testing of its psychometric properties was conducted. The original study thoroughly reported the instrument development process and provided evidence of its validity and reliability in a sample of patients with diabetes [33]. However, its intra-rater reliability was not tested.

The Foot Self-Care Behaviour Questionnaire [34] demonstrated both content validity and homogeneity. Furthermore, the Foot Self-Care Observation Guide [35] reported sufficient evidence of its content validity, internal consistency, test-retest reliability, and feasibility.

### 3.7. Synthesis of the Results

There are several instruments available for measuring foot self-care, all of which especially focus on foot self-care among people with diabetes. However, among the investigated studies, only six instruments were used more than once, thus providing scattered psychometric evidence. In addition, although the instruments were tested for validity and reliability, the testing process for the different instruments varied, with their predominant focus being on content validity and homogeneity.

## 4. Discussion

The current study identified six instruments that measure foot self-care, all of which focus primarily on patients with diabetes. Although each instrument had sufficient evidence supporting its usefulness in evaluating foot self-care, they focused mostly on content validity and homogeneity. This indicates the need for systematic and comprehensive psychometric testing of these instruments. In addition, a wealth of instruments used in single studies was found to have limited evidence of their development processes and psychometrics. Moreover, with the increasing volume of tools used for this purpose, duplication and variability were some of the challenges faced in choosing specific instruments among the available ones.

The selected instruments varied in terms of their complexity regarding the items and factors covered. A clear definition of the construct is necessary to search for the most accurate instrument in terms of a given context [54]. In this context, foot self-care is a complex construct that encompasses knowledge (to know what to do and how), skills (how to care for the feet in real life by implementing the correct procedures), and attitude (motivation for carrying out foot self-care). However, since there is currently no standardized definition of foot self-care, the selected instruments were free to measure different kinds of factors related to foot self-care. The main content of the instruments was related to foot inspection, the type of footwear, socks, and the warming of the feet. This is probably because these areas are theoretically relevant and fundamental when it comes to feet care for patients with diabetes. However, in the future, a concept analysis of foot self-care could be beneficial for improving the measurement focus of foot self-care instruments. The NAFF [32] was found to be the most comprehensive instrument covering a wide number of foot self-care activities. Moreover, the target population in the studies was predominantly patients with diabetes. Therefore, constructing an instrument for measuring basic foot self-care activities performed by the general population could be relevant for population-based health promotion programmes.

Although the construct of interest is generally clearly defined when searching for an appropriate instrument, one should be aware of whether development study and psychometric testing of the specific instrument on the target population were performed [54].

In the future, a strong emphasis should be placed on testing construct validity and reliability in terms of an instrument’s intra-rater and test-retest (stability) reliability. A particularly significant need is to conduct further psychometric evaluation studies on existing instruments and adapt them accordingly, focusing especially on the psychometric properties that are rarely evaluated, such as reliability, measurement error, and responsiveness [14,16]. From a clinical perspective, more information on the clinical feasibility of the evaluated instruments is necessary. To address this issue, researchers could benefit from planning a detailed testing procedure for their instruments. Since all the elements of validity and reliability are impossible to test in a single study, collecting data from different settings and participants, such as patients under home care or those having long-term health conditions like rheumatoid arthritis, could help cumulate more evidence and, in turn, improve the methodological quality of the instruments. Moreover, researchers should not rely solely on internal consistency as an indicator of reliability [14].

Given the variety of instruments that are currently available, understanding the quality of evidence about an instrument for evaluating its measurement properties is essential to make an informed selection of the most appropriate tool and properly assess foot self-care in the population of interest. The consistent use of instruments that have been assessed as valid and reliable allows for the systematic and credible monitoring and comparison of measured results [16]. This highlights the need for a clear definition of foot self-care to help focus future research accordingly.

Authors should discuss the results and how they can be interpreted from the perspective of previous studies and of the working hypotheses. The findings and their implications should be discussed in the broadest context possible. Future research directions may also be highlighted.

### Limitations

The results of this review need to be interpreted while also considering some limitations. First, although the literature search on the selected databases was comprehensive, adding more databases would have provided more hits. In addition, the systematic search conducted across the three databases provided several duplicates, indicating overlapping content. To ensure accurate and comprehensive search terms, pilot searches were conducted, and the search terms were modified and approved by the research team. One researcher (MS) conducted the search, while the study selection was handled by two independent researchers (A-MM, MS). No discrepancies that needed intervention by a third researcher evaluation were encountered. Moreover, data retrieval and analysis were conducted by two researchers (JS, MS) to ensure the transparency of the analysis.

This review was limited to studies published in English. As a result, studies published in other languages that deal with the development of good-quality instruments or the measurement properties of the selected studies were omitted. Therefore, further research can focus on studies conducted in other languages. Moreover, the selected framework developed by Zwakhalen and colleagues [28] represents a general structure for evaluating the psychometric qualities of health measurement scales.

## 5. Conclusions

Although many instruments were identified as potentially suitable for evaluating foot self-care, deficits in demonstrating adequate measurement properties were recognized across all the domains of reliability and validity. Particularly, the evidence of the instruments’ sensitivity to detecting changes in foot self-care is required before they are used as outcome instruments, such as for interventions.

The number of items evaluated in the selected studies ranged from 4 to 29. This supports the need for a concise instrument to minimize patient burden, maximize patient engagement, and ensure the collection of meaningful data [16].

A considerable variety of instruments are used to measure foot self-care, with a small proportion being used consistently. Moreover, substantial variability exists in their level of methodological rigour. Foot self-care instruments are important indicators of patients’ competence in promoting and maintaining their foot health. With precise, well-targeted, and sensitive instruments, health care professionals may be able to monitor the progress of their patients’ foot self-care and evaluate the impact of educational foot health interventions [14]. In addition, in terms of their clinical utility, foot self-care instruments are important for enhancing patients’ engagement, outcome evaluation, and the evaluation of their motivation for carrying out foot health care. Future research should focus on testing the psychometric properties of the instruments as it could provide the benefit of incorporating tests from modern test theory, such as Rasch analysis. Most importantly, researchers and clinicians can take recourse to the information provided in this review to make informed choices when selecting an instrument for their purposes.

## Figures and Tables

**Figure 1 jpm-13-00434-f001:**
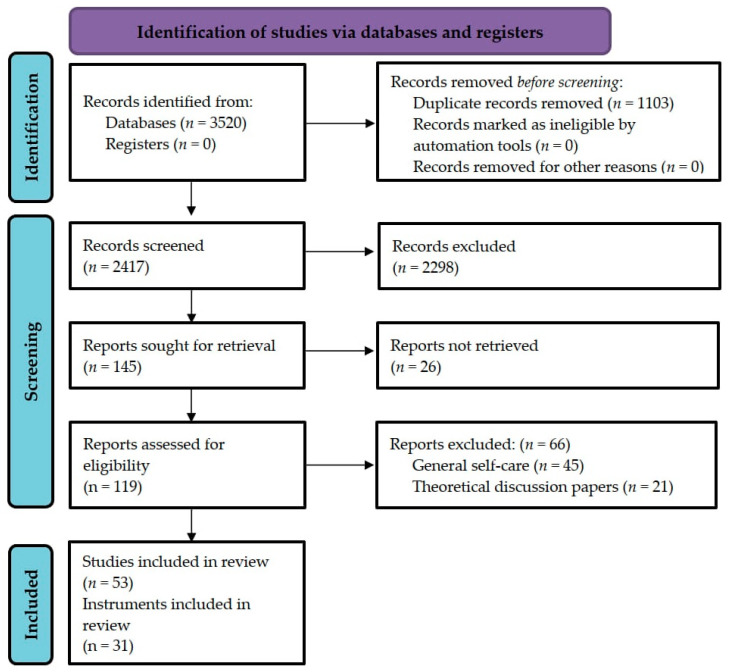
Flowchart of the study selection process.

**Table 1 jpm-13-00434-t001:** Content and number of items in the instruments measuring foot self-care.

Name of Instrument, Reference	Number of Items	Instrument Content												
		Foot Inspection	Foot Wash	Foot Soaking	Drying the Feet	Skin Care	Skin Moisturization	Nail Cutting	Foot Warming	Footwear Assessment	Selection and Type of Footwear	Socks	Walking Barefoot	Attitude toward Foot Self-Care
Diabetes Foot Self-Care Behavior Scale (DFSBS) [30]	7	2	1		1		1			2				
Summary of Diabetes Self e Care Activities (SDSCA) [31]	4	1		1	1					1				
Nottingham Assessment of Functional Footcare (NAFF) [32]	29	1	1		2	3	2	1	4	2	7	4	2	
Diabetic foot self-care questionnaire of the University of Malaga, Spain (DFSQ-UMA) [33]	15	2			2	1		1	1		4	2		2
Foot Self-Care Behaviour (FSCB) questionnaire [34]	17	1	1			2	1	1	2	1	4	2	2	
The Foot Self-Care Observation Guide (FSCOG) [35]	16	3	1		1	3	1	1		2	1	3	1	
number of items/content		10	4	1	7	9	5	4	7	8	16	11	5	2

**Table 2 jpm-13-00434-t002:** Psychometric evidence of six instruments measuring foot self-care, analysed against the criteria proposed by Zwakhalen and colleagues [28].

Instrument	Origin of Items	Number of Participants	Validity					Reliability			
			Content	Criterion	Construct I: in Relation to Other Tests	Construct II: Differentiates	Homogeneity	Intra-Rater	Test-Retest	Feasibility	Total Score
**Diabetes foot self-care behavior scale (DFSBS)**											
Diabetes foot self-care behavior scale (DFSBS) [30]	2	2	2	2	2	2	2	0	0	2	16
Use of DFSBS in Iran [36]	2	2	2	0	0	0	2	0	0	0	8
Use of DFSCBS in Taiwan [37]	2	2	0	0	0	0	2	0	0	0	6
Use of DFSBS in State of Palestine [38]	2	2	2	0	0	0	2	0	2	0	10
Use of DFSBS in Malaysia [39]	2	2	2	0	0	0	0	0	0	0	6
Use of DFSBS in Turkey [40]	2	2	0	0	0	0	1	0	0	0	5
Use of DFSBS in Malaysia [41]	2	2	2	0	0	0	1	0	0	2	9
**The Summary of Diabetes Self-Care Activities (SDSCA): foot care**											
Use of SDSCA in United States [42]	2	2	0	0	0	0	0	0	0	0	4
Use of SDSCA in Tanzania [12]	2	2	0	0	0	0	0	0	0	0	4
Use of SDSCA in Brazil [43]	0	2	0	0	0	0	0	0	0	0	2
Use of SDSCA in Filippines [44]	2	2	0	0	0	0	0	0	0	0	4
Use of SDSCA in South Africa [45]	2	2	0	0	0	0	0	0	0	2	6
The Diabetes Self-Care Activities Questionnaire (DSQ) [46]	2	2	0	0	0	0	0	0	0	0	4
**Nottingham Assessment of Functional Foot-Care questionnaire (NAFF)**											
The Nottingham Assessment of Functional Footcare, original study [32]	2	2	0	0	0	2	1	0	2	2	11
The use of NAFF in United Kingdom [47]	2	2	0	0	0	2	1	0	2	0	9
The use of NAFF in United Kingdom [48]	2	2	0	0	0	0	0	0	0	0	4
The use of NAFF in United Kingdom [49]	2	2	2	0	0	0	2	0	0	0	8
**Diabetic foot self-care questionnaire of the University of Malaga, Spain (DFSQ-UMA)**											
Diabetic foot self-care questionnaire of the University of Malaga, Spain (DFSQ-UMA) [33]	2	2	2	2	2	2	2	0	2	2	18
The use of DFSQ-UMA in Spain [50]	2	2	0	0	0	0	0	0	0	0	4
**Foot Self-Care Behaviour (FSCB) questionnaire**											
The use of FSCB in Australia [51]	2	2	2	0	0	0	2	0	0	0	8
The use of FSCB in Australia [52]	2	2	0	0	0	0	0	0	0	0	4
**The Foot Self-Care Observation Guide**											
Original study [35]	2	2	1	0	0	0	1	0	2	2	10
Gökdeniz & Sahin 2020 [53]	2	2	0	0	0	0	2	0	0	0	6

## Data Availability

The data presented in this study are available on request from the corresponding author.

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
