# Peer review of "How to Measure Foot Self-Care? A Methodological Review of Instruments"

_jpm, 2023, doi:10.3390/jpm13030434_

Round 1

Reviewer 1 Report

REVIEW of jpm-2213107

How to measure foot self-care? A methodological review of instruments

General comments:

This is about the review of foot self-care in three international scientific databases—

Medline (PubMed), CINAHL (Ebsco) and Embase. It is interesting and helpful for measuring foot self-care. At present, software and hardware instruments of foot self-care are more, but there is no uniform standard. For example, what is foot pressure of the normal standard? The hardware instrument is only to measure the foot pressure. They didn't provide a comparison standard. The result of foot self-care measurement is used to diagnose foot illnesses with patients. This will mislead the rehabilitation technician. Especially in the middle of learning foot care personnel, it is easy to follow the crowd. For this review, some comments are as follows:

Abstract:

Line 18: “A total of 31 instruments were identified, of which six were observed to have been used more than once”. Which six? What is the consistency of the six instruments? This will be better than the “Line 19-20” of following expression.

Line 22-25: What parameters authors should list the analyzing content of foot self-care?

Introduction:

The first paragraph in introduction section is really weird

Line 40-44: Foot problems are very common. It involves the several groups of people. Such as the flatfoot, pronation of forefoot, valgus of heel, older people, diabetes mellitus. Authors should check the more references.

Line 47-51: Fonts are different from others.

Line 45-69: This was puzzled me about the content. What does the author mean by foot self-care? Authors showed the “foot self-care activities of patients with diabetes” “foot ulcers” in Line 67, but there are no other foot illness problems (flatfoot, pronation of forefoot, valgus of heel) .

Line 71-84: The author has been talking about the theoretical content, but has not touched on the specific method of what kind of foot care needs.

Line 85-97: Authors showed ‘The validity and reliability of foot- and ankle-related instruments have increasingly been under investigation. However, the target of most studies has been the assessment of foot and ankle symptoms and functions rather than foot self-care’, many researchers are also concerned about the functional recovery of the foot. However, I don't understand and get what are the parameters of foot self-care. At least it was not in the introduction. Is it just theory?

Materials and Methods:

Line 127: In the ‘2) included an instrument that measures foot self-care (subjective or objective), 3) provided evidence of psychometric properties of the foot self-care instrument’, Are there any subjective scales? Are there any objective parameters? What is the evidence of psychometric properties?

Line 219: what is the foot self-care content?

Results:

From the results, authors showed the diabetes foot self-care and nothing else foot illness problems. Is the title of the paper appropriate? In Table 1, ‘Measurement aim’ is so much without uniformity. ‘Number of items, subscales/parts’ did not show the content of items. It was confused me.

Line 244: although authors showed ‘detailed description of the six instruments’, what parameters did you investigate? It was only ‘checking the bottom of the feet and between toes, washing between toes, drying between toes after washing….’, how to objectively assess ‘frequency’? What is the sample table or standard of the total score? what parameters are included? Therefore, the results sections are always confused me. For example, ‘16 items (8 for knowledge, 8 for practice)’ in Table 1, what is ‘8 for knowledge, 8 for practice’ ?.....

Discussion:

Since this is a review of foot self-care, it should cover foot problems of the basics, such as flatfoot, valgus of heel, uneven foot pressure…

Line 356-364: Psychological test and psychological scale included two or three hundred methods, which authors choose?

Line 371: if the instruments did not show the some output parameters, what did it show?

Author Response

Dear Reviewer,

Thank you for your comments. We have taken them all in to consideration and revised the article. Unfortunately some of the comments were hard to understand, but we have tried our best to follow your idea of revision. In the attachment we provide point-by-point responses to your comments. We are willing to revise our article further.

Yours sincerely,

Authors.

---------------------------------------

REVIEW of jpm-2213107

How to measure foot self-care? A methodological review of instruments

 General comments:

This is about the review of foot self-care in three international scientific databases—

Medline (PubMed), CINAHL (Ebsco) and Embase. It is interesting and helpful for measuring foot self-care. At present, software and hardware instruments of foot self-care are more, but there is no uniform standard. For example, what is foot pressure of the normal standard? The hardware instrument is only to measure the foot pressure. They didn't provide a comparison standard. The result of foot self-care measurement is used to diagnose foot illnesses with patients. This will mislead the rehabilitation technician. Especially in the middle of learning foot care personnel, it is easy to follow the crowd. For this review, some comments are as follows:

                We are sorry but we did not understood what you mean with this comment. Assessment of foot pressure is a podiatric method. It is a method that requires professional competence. In our study (review) we are interested of instruments (more clearly said questionnaires) that focus on evaluation of foot self-care, meaning the activities what a patient themselves do to care for their feet. Results of foot self-care measurements are not used to diagnose any foot illness, instead they provide a podiatrist or clinician understanding if patient is caring for own feet and what way and how often.

Abstract:

Line 18: “A total of 31 instruments were identified, of which six were observed to have been used more than once”. Which six? What is the consistency of the six instruments? This will be better than the “Line 19-20” of following expression.

                Thank you for this comment. Due to limited number of words in the abstract, we have not listed these six instruments. The names of these instruments can be found from the text under subtitle 3.3. Description of analysed instruments. We were unsure about your comment related to consistency, sorry. In the abstract we only mention that despite a rather high number of instruments identified in the foot self-care, only few of them are used consistently, and we mean that only these six instruments seem to be used in research more often than the other ones.

 Line 22-25: What parameters authors should list the analyzing content of foot self-care?

                We are sorry, but we have problems to understand this comment. In lines 22-25 we report results of analysing instruments’ content validity. We are not reporting in this sentence any content of foot self-care. In the introduction we have clarified what we mean with foot self-care in this paper.

Introduction:

The first paragraph in introduction section is really weird

                This was an error, and the first paragraph is deleted.                                                                                                                                                                                               

Line 40-44: Foot problems are very common. It involves the several groups of people. Such as the flatfoot, pronation of forefoot, valgus of heel, older people, diabetes mellitus. Authors should check the more references.

                Yes, we agree that foot problems are very common affecting many patient groups. We have added some more references to highlight the foot problems.

Line 47-51: Fonts are different from others.

                Fonts are revised.

Line 45-69: This was puzzled me about the content. What does the author mean by foot self-care? Authors showed the “foot self-care activities of patients with diabetes” “foot ulcers” in Line 67, but there are no other foot illness problems (flatfoot, pronation of forefoot, valgus of heel) .

                We have clarified this in the text. Foot self-care in this study means activities that patients are doing to promote their foot health or care for foot problems. We are not limiting this to any patient group, instead we are focusing on all adult patients.

Line 71-84: The author has been talking about the theoretical content, but has not touched on the specific method of what kind of foot care needs.

                Thank you. We have clarified this in the text.

Line 85-97: Authors showed ‘The validity and reliability of foot- and ankle-related instruments have increasingly been under investigation. However, the target of most studies has been the assessment of foot and ankle symptoms and functions rather than foot self-care’, many researchers are also concerned about the functional recovery of the foot. However, I don't understand and get what are the parameters of foot self-care. At least it was not in the introduction. Is it just theory?

                We have added to the introduction the parameters of foot self-care. Parameters of foot self care include foot inspection, skin and nail care, care for foot structural deformities (e.g. hammer toes), pain management and wearing appropriate footwear.

Materials and Methods:

Line 127: In the ‘2) included an instrument that measures foot self-care (subjective or objective), 3) provided evidence of psychometric properties of the foot self-care instrument’, Are there any subjective scales? Are there any objective parameters? What is the evidence of psychometric properties?

                All instruments that we identified in this review were subjective, meaning that patients reported how and how often they performed foot self-care. We did not find any objective instruments, such as observation of foot self-care. Evidence of psychometric properties mean that the studies to be included in the review needed to report at least some evidence of psychometrics of the instrument (meaning evidence of validity and/or reliability).

Line 219: what is the foot self-care content?

                We have clarified this into the text.

Results:

From the results, authors showed the diabetes foot self-care and nothing else foot illness problems. Is the title of the paper appropriate? In Table 1, ‘Measurement aim’ is so much without uniformity. ‘Number of items, subscales/parts’ did not show the content of items. It was confused me.

                The title of the paper is in line with the aim of our review. We started to seek what foot self-care instruments there are, not limiting our interest to any specific patient group. As a result. we identified that all instruments that we included focused on patients with diabetes. We will not modify the title, as the modification will change the original idea of the entire review. Instead, in the discussion we are trying to highlight the need to widen the foot self-care evaluation from patients with diabetes to other patient groups as well.

Line 244: although authors showed ‘detailed description of the six instruments’, what parameters did you investigate? It was only ‘checking the bottom of the feet and between toes, washing between toes, drying between toes after washing….’, how to objectively assess ‘frequency’? What is the sample table or standard of the total score? what parameters are included? Therefore, the results sections are always confused me. For example, ‘16 items (8 for knowledge, 8 for practice)’ in Table 1, what is ‘8 for knowledge, 8 for practice’ ?.....

                We are sorry that our paper confused you. Detailed description of the six instruments include description of what the instrument measures, how many items it has, how the items are divided or grouped, what the response options are and how the results are interpreted. This is typical way to describe the instrument’s descriptive content.

Discussion:

Since this is a review of foot self-care, it should cover foot problems of the basics, such as flatfoot, valgus of heel, uneven foot pressure…

                This is a review of instruments measuring foot self-care. We are not interested of foot-self care as a topic. We have added a mention that foot self-care is important in caring for different foot disorders, such as flatfoot, hallux valgus and metatarsalgia.

Line 356-364: Psychological test and psychological scale included two or three hundred methods, which authors choose?

                We am sorry, but we have problems to understand this comment. We have not discussed or reported about psychological tests or scales, instead we are focusing on discussing the psychometric properties/evidence of foot self-care instruments.

Line 371: if the instruments did not show the some output parameters, what did it show?

                Instruments’ psychometric evidence was predominantly focused on describing content validity and homogeneity (e.g. internal consistency).

Reviewer 2 Report

Dear authors,

The aim of the current study is to describe and assess from the methodological perspective the available feet self-care questionnaires existing in the literature. The introduction offers strong arguments which sustain the motivation of this review. However, I have the following remarks related to this section:

-the first paragraph isn't related to the content of the proposed review and represent general suggestions for authors. I suggest to remove this paragraph.

-in the third paragraph (lines 45-62) the  authors realize an objective definition of the term feet self-care starting from the WHO definition of self-care. In my opinion, there are recurring ideas in this section and I suggest to reformulate it in order to be more more concise and persuasive.

The methodology of the study is well described  providing the eligibility criteria,  the search sentence and the process of  selection of the studies. The authors chose the framework developed by Zwakhalen  to realize an in-depth evaluation according with the steps of the psychometric validation.

In the result section, in table 2 the last row represent the sum of the number of items included in each questionnaire to describe a certain aspect of feet care. I would suggest to justify the reason for you choose this option in the description section. In the third row of the same table you wrote 2x (DFSBS- footwear assessment).  The same suggestion is for the last column of the table 3, relating to the  reported mean.

Author Response

We thank yours and reviewers’ valuable comments to our manuscript. We have taken them all into consideration and revised the manuscript accordingly. All changes are marked with Track changes in the text and as appendix a point-by-point responses to each revision note are provided below.

We are looking forward to continue working with this manuscript and revise it along with your possible comments.

Yours sincerely,

the Authors.

---------------------------------

Reviewer 1.

Dear authors, The aim of the current study is to describe and assess from the methodological perspective the available feet self-care questionnaires existing in the literature.

The introduction offers strong arguments which sustain the motivation of this review. However, I have the following remarks related to this section:

-the first paragraph isn't related to the content of the proposed review and represent general suggestions for authors. I suggest to remove this paragraph.

                Thank you for pointing this out. We are sorry that was our mistake due to oversight. The first       paragraph is removed.

-in the third paragraph (lines 45-62) the authors realize an objective definition of the term feet self-care starting from the WHO definition of self-care. In my opinion, there are recurring ideas in this section and I suggest to reformulate it in order to be more more concise and persuasive.

                We have clarified this paragraph, removed some sentences and reformulated the definition         of foot self-care and competence-related aspects.

The methodology of the study is well described providing the eligibility criteria, the search sentence and the process of selection of the studies. The authors chose the framework developed by Zwakhalen to realize an in-depth evaluation according with the steps of the psychometric validation. In the result section, in table 2 the last row represent the sum of the number of items included in each questionnaire to describe a certain aspect of feet care. I would suggest to justify the reason for you choose this option in the description section.

                We have added description and justification for summing the number of items included in             each questionnaire.

In the third row of the same table you wrote 2x (DFSBSfootwear assessment).

                Thank you for noticing this. “x” is wrong and is now deleted. The box should include only                number 2.

The same suggestion is for the last column of the table 3, relating to the reported mean.

                We have added description and justification for summing up the number representing    psychometric evidence.

Round 2

Reviewer 1 Report

Please view the uploaded Word file

Author Response

Thank you for your comments. Please see a separate Word file of our responses.

Reviewer 2 Report

Dear authors, thank you for your diligent effort to enhance the manuscript submitted. 

It is my opinion that tables can prove ardous to read due to their lengh and structure. As a solution, relocating them to a separate file or restructuring their format may prove beneficial. 

Sincerely yours,

Alina Popa MD, PhD

Author Response

Thank you for your comments. We have made a separate Word file to explain our revisions.
